# Optimization of Novel Human Acellular Dermal Dressing Sterilization for Routine Use in Clinical Practice

**DOI:** 10.3390/ijms22168467

**Published:** 2021-08-06

**Authors:** Hanna Lewandowska, Andrzej Eljaszewicz, Izabela Poplawska, Marlena Tynecka, Alicja Walewska, Kamil Grubczak, Jordan Holl, Hady Razak Hady, Slawomir Lech Czaban, Joanna Reszec, Grażyna Przybytniak, Wojciech Głuszewski, Jarosław Sadło, Małgorzata Dąbrowska-Gralak, Cezary Kowalewski, Piotr Fiedor, Tomasz Oldak, Artur Kaminski, Zbigniew Zimek, Marcin Moniuszko

**Affiliations:** 1Division of Radiation Modified Polymers, Centre for Radiation Research and Technology Institute of Nuclear Chemistry and Technology 16 Dorodna St., 03-195 Warsaw, Poland; h.lewandowska@ichtj.waw.pl (H.L.); g.przybytniak@ichtj.waw.pl (G.P.); w.gluszewski@ichtj.waw.pl (W.G.); j.sadlo@ichtj.waw.pl (J.S.); m.dabrowska@ichtj.waw.pl (M.D.-G.); 2Department of Regenerative Medicine and Immune Regulation, Medical University of Białystok, Waszyngtona 13, 15-269 Białystok, Poland; andrzej.eljaszewicz@umb.edu.pl (A.E.); marlena.tynecka@umb.edu.pl (M.T.); alicja.walewska.95@gmail.com (A.W.); kamil.grubczak@umb.edu.pl (K.G.); jordanm.holl@umb.edu.pl (J.H.); 3Department of Medical Pathomorphology, Medical University of Białystok, 15-230 Białystok, Poland; iptaborda@gmail.com (I.P.); joannareszec@gmail.com (J.R.); 4Department of General and Endocrinological Surgery, Medical University of Białystok, 15-276 Białystok, Poland; hadyrazakh@wp.pl; 5Department of Anaesthesiology and Intensive Therapy, Faculty of Health Sciences, Medical University of Białystok, 15-274 Białystok, Poland; slawomir.czaban@umb.edu.pl; 6Department of Dermatology and Immunodermatology, Medical University of Warsaw, 02-091 Warsaw, Poland; ckowalewski@wum.edu.pl; 7Department of General and Transplantation Surgery, Medical University of Warsaw, 02-091 Warsaw, Poland; piotrfiedor@wp.pl; 8Polish Stem Cell Bank (PBKM), 00-867 Warsaw, Poland; Tomasz.Oldak@pbkm.pl; 9Department of Transplantology and Central Tissue Bank, Medical University of Warsaw, 02-091 Warsaw, Poland; artur.kaminski@wum.edu.pl; 10Department of Allergology and Internal Medicine, Medical University of Białystok, 15-276 Białystok, Poland

**Keywords:** human acellular dermal matrices, ADM, biological dressing, skin substitutes, radiation sterilization, collagen

## Abstract

Gamma rays and electrons with kinetic energy up to 10 MeV are routinely used to sterilize biomaterials. To date, the effects of irradiation upon human acellular dermal matrices (hADMs) remain to be fully elucidated. The optimal irradiation dosage remains a critical parameter affecting the final product structure and, by extension, its therapeutic potential. ADM slides were prepared by various digestion methods. The influence of various doses of radiation sterilization using a high-energy electron beam on the structure of collagen, the formation of free radicals and immune responses to non-irradiated (native) and irradiated hADM was investigated. The study of the structure changes was carried out using the following methods: immunohistology, immunoblotting, and electron paramagnetic resonance (EPR) spectroscopy. It was shown that radiation sterilization did not change the architecture and three-dimensional structure of hADM; however, it significantly influenced the degradation of collagen fibers and induced the production of free radicals in a dose-dependent manner. More importantly, the observed effects did not disrupt the therapeutic potential of the new transplants. Therefore, radiation sterilization at a dose of 35kGy can ensure high sterility of the dressing while maintaining its therapeutic potential.

## 1. Introduction

Recently, much effort has been made to find an effective treatment option for patients suffering from skin injuries and hard-healing wounds such as burns, ulcers, epidermolysis bullosa, and others. Despite significant progress in the fundamental mechanisms regulating wound healing and the contribution of skin substitutes in this regard, effective treatment options for the vast majority of hard-healing wounds are still limited to invasive autografts and allografts. Furthermore, the use of autologous skin grafts is greatly limited by size and availability [1], as well as the harm it may cause in individuals with abnormal wound healing. A commonly considered and established alternative is the use of acellular dermal matrices (ADMs), which meet most clinical expectations and may serve as a dressing alone or in combination with cellular therapy [1,2]. In fact, the use of acellular dermal scaffolds as stem cell carriers represents an exciting option to improve wound closure, reepithelization, and wound remodeling [1,3,4,5].

One crucial element in the production of ADMs from full skin is their necessity to either be produced in aseptic conditions, or be subjected to an additional sterilization procedure. A variety of methods have been proposed as an effective method for ADM sterilization, including brief immersions in ethanol or peracetic acid, gamma or ultraviolet irradiation, ethylene-oxide exposures, and the use of disinfectants, such as antibiotic, antimycotics, or bacteriostatic agents, such as sodium azide [6]. Currently, radiation sterilization is proposed as an effective way to ensure end-product sterility in biologically derived materials and products [7,8].

Sterilization aims to destroy and/or remove all contaminating micro-organisms from a given material or product. For implantable materials, the recommended method of sterilization is via ionizing radiation [9]. The application of ionizing radiation results in its subsequent absorption by viral and micro-organism contaminants, consequently inducing the excitation of atoms and molecules. This triggers a chain of secondary biological reactions, causing damage to key intracellular structures and molecules [10]. Notably, there are two different sources of radiation used in sterilization procedures, namely gamma sources (with predominantly used cobalt-60) and electrical sources (based on accelerators that provide electron beams).

The advantages of radiation sterilization include the possibility of sterilizing objects already in sealed packaging, which eliminates the risk of secondary contaminations due to downstream packing. In addition, during radiation sterilization, no high temperature is required, with sterilization further able to be performed on dry ice to reduce the likelihood of product degradation [7].

The choice of sterilization dose remains a critical parameter affecting final product structure and functionality [7,11]. According to the International Atomic Energy Agency (IAEA), the minimum dose for tissue sterilization should reach or exceed 25 kGy. However, to ensure the microbiological sterility of biomaterials, higher doses are often considered. Notably, the use of radiation sterilization requires optimization to minimize its negative influence on the biomaterial structure. Usually, a dose of 35 kGy is considered sufficient to ensure septicity and maintain satisfactory product functionality [7]. In fact, radiation sterilization of collagen structures may significantly modify its structure by fragmentation, degradation, oxidation, loss of crystallinity or cross-linking [7,12], changes in physicochemical properties, increased solubility, fragmentation of the peptide groups leading to the formation of N-terminated amino and carbonyl groups [13], and cause differences in the composition of amino acids when compared to native fibers [14,15].

To date, the effects of irradiation of human ADMs (hADMs) remain to be fully elucidated. In order to understand the effect of ionizing radiation on proteins, it is necessary to control experimental conditions, especially temperature and phase modulation. Irradiation of a sample containing water at room temperature produces free radicals from the solvent (water radiolysis products) as the main primary products. These radicals then react chemically with proteins, leading to secondary processes such as cleavage of the peptide backbone. In some cases, these cleaved peptide fragments can aggregate [16], cross-linking proteins [17]. Gamma rays and electrons with kinetic energy up to 10 MeV are routinely used to sterilize biomaterials.

Several major conclusions can be drawn from studies conducted over the past 20 years: First, the magnitude and nature of direct effects are often a consequence of the dose-dependent application of irradiation. Second, the ionization of atoms or groups within individual proteins results in radicals that are involved in the loss of side chains and the cleavage of the backbone, as well as in cross-linking. Third, the type and strength of ionizing radiation play a crucial role. The plausible mechanisms and molecular outcomes of ionizing radiation on proteins have been thoroughly examined in [18].

Given the demonstrable effect of irradiation on proteins, irradiation-induced modifications of collagen fibers may change ADM properties and subsequently influence therapeutic potential. Currently available ADMs are manufactured from human cadaveric skin, while others are prepared from xenogeneic skin fragments. Here, we present a novel approach in which abdominoplasty skin was used for ADM preparation as a part of a novel BIOOPA dressing, which was successful for epidermolysis bullosa treatment [3,5]. We evaluate the effects of different doses of radiation sterilization by high-energy electron beams on the collagen structure, free radicals’ development, and cellular immune responses to novel ADM prepared from human abdominoplasty skin (BIOOPA scaffold). In fact, the experiments were designed for the optimization of the BIOOPA scaffold sterilization protocol for routine use in clinical practice.

## 2. Results

### 2.1. Radiation Sterilization Allows for the Preservation of hADM Structure but Induces Collagen Fiber Degradation, Fragmentation, and Release

First, we aimed to evaluate the effects of different doses of radiation sterilization on collagen structure (Figure 1). By using Masson’s trichrome staining, we found that the general structure of irradiated hADM remains stable and comparable with its native (non-irradiated, 0 kGy) counterparts. The changes in the fiber structure and architecture of scaffolds after radiation are imperceptible. However, as expected, collagen fiber fragmentation was observed in irradiated ADMs. It was confirmed in SDS-PAGE electrophoresis after pepsin digestion (see further in the text, Figure 2A). Next, we analyzed irradiation effects on primary collagen fibers and vitronectin, constituting the essential structural components of hADMs (Figure 1). Additionally, no significant differences in collagen different collagen and vitronectin architectures were observed (Figure 1).

Pepsin digestion of the telopeptides in hADM collagen revealed individual fractions of collagen and the contribution of their alpha, beta and gamma chains both before and after irradiation (Figure 2). Gel electrophoresis results showed the beta chains and gamma chains being seen above the alpha chains of collagen (beta and gamma chains consist of parts of the triple helix that were not totally separated). The alpha chains (1 and 2) display bands below the 98 kilodalton (KDa) Molecular Weight (MW) marker, beta chains around 209 KDa MW, and gamma chains appearing as the heaviest and, therefore, slowest bands [19]. More importantly, the individual strands tend to dissipate as irradiation dose increases (Figure 2A). In summary, a marked difference in the contribution of individual protein fractions could be seen, accompanied by a massive increase in the band background (smearing of the bands) consistent with an increase in radiation dosage.

Targeted analysis of hADM lysates showed cleavage of collagens I, III, and IV but not collagen VII fibers after skin processing (Figure 2B). Moreover, we observed increased fragmentation and degradation of collagens I and III after radiation sterilization. Next, we evaluated collagen release from non-irradiated (0 kGy) and irradiated hADMs. Interestingly, we found that although collagen I and VII were released from irradiated hADMs, collagen III was detected in both irradiated and non-irradiated hADM samples. In addition, radiation sterilization was clearly observed to induce degradation of collagens I and VII. Importantly we confirmed collagen I fragmentation and release is increased in a radiation dose-dependent manner (Figure 2B). Similar to collagen I, collagen VII degradation is readily induced by the sterilization process. However, in contrast to this, we found no association between collagen VII release levels and irradiation dosage.

### 2.2. Human Acellular Dermal Matrix Irradiation Induces Free Radicals’ Release

Having established that the sterilization process induces collagen fiber fragmentation and degradation, we attempted to analyze the molecular mechanisms underlying the irradiation-induced degradation of collagen fibers. In this regard, we investigated the production of free radicals as a consequence of irradiation by examining electron paramagnetic resonance (EPR). The irradiation of samples at a low temperature (77 K) enabled the observation of the mechanisms that were stochastically generated by the ionizing radiation radicals that dissipated towards the most energetically favorable states. Using this information, it was possible to hypothesize plausible mechanisms underpinning previously-witnessed collagen fiber fragmentation.

Figure 3A presents spectra recorded at lower microwave power. The absorption lines are relatively wide, and the entire recorded spectrum is asymmetrical, indicating that at least some radicals demonstrate anisotropic interactions. The spectra are complex, with at least three individual signals that can be distinguished and characterized by simulation (Figure 3C). The most intensive pattern, overlapped with an unknown broad singlet, consisted of five lines. The hyperfine splitting was 1.95 mT, and the g-factor was 2.0031, suggesting that the signal can be attributed to the secondary alkyl radical –CH_2_-^•^CH_2_, in which unpaired spin interacts with four equivalent protons. The EPR spectra also show an asymmetric singlet. This singlet is especially pronounced at 10 mW microwave power (Figure 3B), i.e., when saturation effects suppress the contribution of the alkyl radical-related signal in the EPR spectrum, which was confirmed in preliminary studies. The intensity of the singlet increased with the increase in the annealing temperature (Figure 3), up to 220 K. The spectrum simulation allowed to estimate the following spectral parameters: g_1_ = 2.0021, g_2_ = 2.0083 and g_3_ = 2.0310 (Figure 3C). Notably, these values are similar to those found in other biopolymers and plastics [20] for peroxyl radicals.

Simulations of all identified radicals are presented in Figure 3C. The presence of peroxyl radicals was confirmed even when hADMs were irradiated at the lowest temperatures that were tested.

Furthermore, the EPR spectra presented in Figure 3A,B were double integrated, ultimately revealing a decrease in the number of radicals generated during thermal annealing of the sample (Figure 4A). It was assumed that the content of radicals at 100 Kelvin (K) was 100%, and the decrease in their frequency was calculated accordingly to this assumption. Ultimately, the relative concentration of paramagnetic species gradually decreased due to their irreversible recombination.

In order to determine the stability of collagen radicals during storage on dry ice, the grafts were irradiated with a dose of 35 kGy at 194.7 K, and then the EPR spectra were periodically measured while maintaining this temperature (Figure 4B). The radical decay profiles were fitted using second-order decay functions. Although half of the radicals’ total population disappeared within 6 h of irradiation, their presence could still be detected even after 38 h.

The ADM samples irradiated with the increasing doses of e-beam radiation, i.e., 0, 15, 25, and 35 kGy, were measured for the irradiation-induced hydrogen production and oxygen consumption. The values of the H_2_ production and O_2_ consumption displayed a linear dependence on the irradiation dose. The radiation yield of hydrogen emission G(H_2_) determined based on the GC experiments was 0.026 µM H_2_/J, while the radiation yield of oxygen consumption G(O_2_) was 4.6 times higher (0.12 µM O_2_/J). The results are presented in Appendix A.

### 2.3. Different Doses of Radiation Sterilization Do Not Change the Immunogenicity of hADM

Finally, we aimed to assess whether sterilization-induced changes in collagen fibers and free radicals’ generation may affect hADM immunomodulatory potential. Interestingly we found that both native (non-irradiated, 0 kGy) and irradiated hADMs induced the activation of co-incubated immune cells after 5 days. This was manifested by the increased production of inflammatory mediators such as proinflammatory interferon gamma (IFNγ), interleukin (IL)-17, IL-22, IL-1β, tumor necrosis factor (TNF), IL-17, and anti-inflammatory IL-35 (Figure 5), while the levels of IL-4, tumor growth factor beta (TGFβ), and IL-10 appeared to be below detection range (data not shown). Surprisingly, we found no differences in analyzed cytokine levels in cell culture supernatants between native (0 kGy) and irradiated hADMs, showing that radiation sterilization did neither affect nor impair hADM immunomodulatory properties.

Interestingly, the levels of IL-1β, IL-22, and IL-35 were comparable after five days of co-incubation with either hADM or skin. However, unprocessed skin induced higher levels of IL-17 when compared to both irradiated and non-irradiated scaffolds. Moreover, TNF levels were higher in hADM co-incubated cells versus unstimulated controls.

## 3. Discussion

Here, we showed that radiation sterilization of the novel ADM manufactured from human abdominoplasty skin does not alter collagen architecture. The decellularization process allows removing all cellular components of the skin. Consequently, the 3D structure of the extracellular matrix (ECM) of the skin is preserved and may be subjected to further processing. In fact, collagens represent a principal component of ECM of the skin (70–80% of the dry weight of the skin), and therefore in this report, we decided to focus on collagen structure as it represents the major component of our novel ADM, which is most sensitive to irradiation sterilization and allows keeping the design of the scaffold. The collagen remains stable upon irradiation, probably due to its unique macromolecular and intermolecular structure, containing numerous cross-links. The high stability of collagen architecture in response to radiation gives good indications for radiation sterilization of collagen-based medical products. Nevertheless, as expected, irradiation induces fragmentation and degradation of particular collagen fibers, which can be observed after collagen cleavage by pepsin and also in the analysis of free radicals’ development. However, of crucial importance for its therapeutic potential and intended use, we demonstrated that the observed changes do not affect our novel abdominoplasty skin-derived hADM immunogenicity.

Notably, the EPR data allowed for the deduction of mechanisms involving the observed fragmentation and molecular modifications of collagen fibers upon irradiation with 10 MeV EB. As can be seen in Figure 3A, the most intensive pattern in the spectrum of low temperature-irradiated collagens consisted of five lines. The hyperfine splitting was 1.95 mT, and the g-factor was 2.0031, suggesting that the signal can be attributed to the secondary alkyl radical –CH_2_-^•^CH_2_, in which unpaired spin interacts with four equivalent protons. Such an intermediate cannot be formed in the main backbone of collagen but only in the side groups such as lysine, arginine, or hydroxylysine. Based on the spectrum, the localization of unpaired spin cannot be assigned to a specific amino acid residue. Despite this, it appears that the radical forms as a result of C-C or C-N bond scission. It can be inferred that other alkyl radicals must also be consequently generated given the release of hydrogen atoms during irradiation, as confirmed by gas chromatography measurements.

The data indicate that some alkyl radicals produced by scission C-H bonds are rapidly oxidized even under cryogenic conditions [12,13,21]. Consequently, the asymmetric singlet attributed to peroxyl radical RCOO^•^ was observed by EPR spectroscopy. Its spectrum can be easily recognized in a sequence of the signals recorded at 10 mW microwave power (Figure 3B) because, under these conditions, the lines of carbon-centered radicals are partly saturated—in contrast to the anisotropic singlet of peroxy radicals, which increases significantly. Its shape changes with the increase in the intensity of conformational movements associated with the rise of temperature. The peroxyl radical signal increased with temperature, which points at its production up to 250 K. Unfortunately, based on the EPR spectra, it is hard to conclude what kind of radical interconversion was responsible for the production of this peroxyl radical, in large part because no marked decrease in another signal could be seen in conjunction with the observed increase in peroxyl radical signal strength.

Peroxy radicals are able to abstract hydrogen from macromolecules, forming hydroperoxides. This process is not selective but usually involves the exploitation of weak bonds in secondary carbon atoms. Based on the doublet formation at temperatures exceeding 280 K (Figure 3A,C), it appears that glycine residue loses the hydrogen-forming –NH-^•^CH-CO- intermediate. Radical centers localized elsewhere in collagen should demonstrate more lines in EPR spectra due to the presence of β protons at carbon atoms. Hyperfine splitting A(H_α_) = 1.61 mT is slightly lower and the g-factor higher (g = 2.004) than in typical alkyl radicals due to the influence of the neighboring carbonyl group, which confirms the proposed interpretation.

Taken together, these results showed that the oxidation of some radicals occurs very quickly, even when under cryogenic conditions, and that the process is not controlled by O_2_ diffusion due to the sufficient availability of oxygen molecules dissolved in collagen.

In the past, Symons postulated the creation of an R^•^CH_2_ terminal alkyl radical in silk collagen, which resulted from the cleavage of the main chain [22]. However, our recorded spectra did not contain a triplet of hyperfine splitting 2.20 mT associated with its presence. It seems that this type of radical can abstract hydrogen from secondary carbon atoms transforming into herein confirmed by EPR spectroscopy –^•^CH- intermediate, which is thermodynamically favorable. This finding correlates well with the observed scission of collagen backbone.

The decay profiles (Figure 4A) determined for spectra recorded at 10 mT indicated that the peroxy radical was thermally favored versus alkyl radicals and that only part of alkyl radicals is converted to –NH-^•^CH-CO-. The remaining radicals can recombine to form dialkyl peroxides.

The triple helix of collagen has telopeptides implicated in the cross-linking of collagen molecules to form fibrils. Each mature type I collagen molecule has two short non-helical regions at both N- (NTx) and C-terminus (CTx), called telopeptides [23]. The telopeptides possess an uncommon amino acid hydroxylysine (Hyl), which is important for the formation and stabilization of collagen structures due to the action of lysyl oxidase, an enzyme that catalyzes the covalent aldol reaction between the lysine or hydroxylysine residues in the N- and C-telopeptides of adjacent molecules, thus bonding two molecules head-to-tail along the fibril [24]. Pepsin digestion before electrophoresis allows partial cleavage in the telopeptide region, revealing the protofibrils and cleavage of the protofibrils into protofibrillar particles [19]. In the electrophoresis, the alpha chains (1 and 2) display bands that can be herein observed below the 98 K MW marker, with beta chains resting around the 209 K MW and the gamma chains being the slowest migrating band. The bands of undamaged skin collagens are well resolved into these molecular weights, as commonly observed in the literature [19,25,26]. Here, we demonstrated that the chain breakage of the individual collagen chains increased with the dose, as the bands for alpha-, beta- and gamma-chains dissipated. Marked changes in the contribution of individual protein fractions could be seen, accompanied by a massive increase in the band background (smearing of the bands, Figure 2A). Even though these marked changes in the collagen structure (fragmentation) were easily observed via gel electrophoresis after pepsin digestion, they did not manifest into the expected micro- and macrostructural changes of the scaffolds. One of the possible explanations is connected with the structure and properties of fibrils forming the collagen matrix of the hADM, and in particular to the presence of the mentioned telomeric fragments.

Divalent cross-links bind one telopeptide to the helical region of another collagen molecule [27]. The formation of collagen cross-links is associated with the presence of two modified amino acids containing an aldehyde group (such as allysine and hydroxallysine) that react with other amino acids in collagen to form bifunctional, trifunctional, and tetrafunctional cross-links. Cross-linking occurs in a spontaneous, progressive fashion. The cross-links dictate very precise intermolecular alignments in the collagen polymer. The precise pattern of cross-linking is a function of each specific collagen, and the relative abundance of the different cross-links varies between collagen types. [27]. This cross-linking leads to the extended stabilization of the collagen matrix, which maintains its structure even when individual collagen chains are cleaved. Notably, during the healing process and other processes connected to the degradation of the tissue collagen matrix, as in the case of matrix metalloproteinases [28], enzymatic activity leads to digestion of telomeric regions, causing de-cross-linking of the fibers and rendering collagenous chains soluble in water. In these circumstances, unnoticeable macro-scale peptide chain breakage results in the release of short peptide chains, as could be seen in the electrophoresis of irradiated samples of the pepsin-treated ADM collagen.

From the clinical perspective, the primary function of a skin substitute is to support the healing process and re-epithelialization of the impaired wound environment [1,29,30,31]. Notably, wound healing is strictly regulated by complex interactions of different cell types, including immune cells, epithelial cells, endothelial cells, fibroblasts, as well as stem and progenitor cells [32,33,34,35]. Recently, our group members showed that hADMs may increase wound closure and induce epithelization in an experimental mice model of deep wounds and in epidermolysis bullosa patients [5]. Notably, it is well established that three subsequent phases of wound healing—inflammation, proliferation, and remodeling—are orchestrated by immune cells and their related inflammatory mediators [32,33,34,35]. More importantly, the disrupted regulation of pro-inflammatory responses represents one of the hallmarks of delayed wound closure and consequential ulcer development [36]. On the other hand, recent evidence showed that low-grade inflammation is evolutionarily conserved in lower organisms, acting as a crucial trigger mechanism of the wound healing process [37]. However, to date, the contention as to whether clinically useful ADMs should exert immunostimulatory or immunoregulatory properties remains elusive.

However, the sterilization process should not significantly change ADM properties, as this may negatively affect its ability to incorporate into the wound environment during the healing process. We showed here that despite irradiation-induced collagen fiber fragmentation, degradation, and release, as well as the induction of free radicals, the immune-modulatory properties of hADM remain unchanged. Surprisingly, detected inflammatory mediators were relatively high and comparable to those observed in the presence of full skin. Notably, it is well-established that collagen fibers, such as collagen I, collagen III, and collagen IV, induce expression of proinflammatory cytokines, such as IL-1b, IFN-gamma, TNF but not IL-4 and IL-10 [38,39].

All the cytokines mentioned above play a central role in wound healing, with their concentrations shifting with the overlapping phases of wound healing [40]. IL-1β plays an essential role in the healing process’s inflammatory phase and further induces fibroblasts and chondrocytes to collagen production and release, thus supporting extracellular matrix formation and wound remodeling. Importantly, the process of scar formation may be regulated by IFNγ, which has been shown to attenuate excessive scar formation [41,42,43]. Similarly, blocking of IL-10 has been shown to decrease keloid scar formation. The function of fibroblast is also supported by IL-22 signaling. In fact, IL-22 is recognized as one of the crucial players in the early and late wound healing phases in the skin [44]. Furthermore, the inflammatory phase is also supported by TNF and IL-17 family members [45]. TNF is highly conserved and produced predominantly by innate immune cells, namely macrophages [46]. Unfortunately, to date, its role in the following phases of the process remains not clear. In contrast, IL-17 is produced mainly by Th17 cells, thus represents the element of the adaptive immune response. Interestingly, the role of IL-17 is not only limited to the inflammatory phase since IL-17 signaling may act as a trigger of keratinocyte proliferation [47]. On the other hand, persistent and highly-concentrated levels of these cytokines may impair wound closure and, along with inflammatory cells, which promote their secretion, are recognized as one of the reasons for chronic wound and ulcer formation [48]. In summary, our results show that hADMs possess high potential to support wound healing. This ability is not disrupted by the sterilization procedure.

## 4. Materials and Methods

### 4.1. Skin Acquisition and Storage

Human acellular dermal matrices were processed from superficial layers of human skin harvested from bariatric patients undergoing abdominoplastic surgery. Dermatome skin grafts were taken from the skin fold and packed in double-sealed, sterilized foil bags and stored at −80 °C for further processing. All samples were collected following informed consent and upon approval of the Ethics Committee of the Medical University of Bialystok, Poland.

### 4.2. Acellular Dermal Matrix Preparation from Bariatric Skin

Thawed skin grafts were washed 3× in buffered saline (PBS, Corning, Mannasas, VA, USA) and subjected to the decellularization procedure. Briefly, skin grafts were fixed on steel mesh and placed into a sterile container with 1 M NaCl (Sigma Aldrich, St. Louis, MO, USA) supplemented with antibiotics, followed by 24 h incubation with gentle agitation. Next, the epidermis was removed mechanically using tweezers, and dermal layers of the graft were incubated with agitation in a decellularization medium containing (a) PBS with calcium and magnesium ions (Corning, Mannasas, VA, USA), (b) 3% Triton X-100 (Sigma-Aldrich, St. Louis, MO, USA), (c) Deoxyribonuclease I from bovine pancreas 200 KiloUnits/500 mL (SigmaAldrich, St. Louis, MO, USA), (d) RNase T1 ThermoFisher 100,000 Units/500 mL (ThermoFisher, Carlsbad, CA, USA), (e) aprotinin (Sigma-Aldrich, St. Louis, MO, USA), and (f) antibiotics (Sigma-Aldrich, St. Louis, MO, USA). After 24 h incubation, the decellularization buffer was removed, and the graft was submerged in fresh decellularization buffer and incubated for another 24 h with agitation. Next, the decellularization buffer was removed, and the graft was rinsed in fresh ultra-pure filtered water for 4 days, with daily fluid changes. Finally, the skin was lyophilized in an automated −80 °C lyophilizator (LyoQest Telstar) and sealed in double foil bags. Frozen acellular dermal matrices were sent for irradiation to the Institute of Nuclear Chemistry and Technology, Poland.

### 4.3. Irradiation

Frozen hADM in sealed foil bags, provided by the Medical University of Bialystok in thermoboxes filled with dry ice were irradiated with an electron beam (EB) emitted by a linear accelerator installed in a commercial Sterilization Plant at the Institute of Nuclear Chemistry and Technology. Depending on further research, biomaterials absorbed doses of 15, 25, and 35 kGy. The samples were exposed to irradiation at 194.7 K in an air atmosphere. The ADMs were hermetically sealed in double foil bags and placed in a Styrofoam thermobox on the dry-ice layer so that the dry ice density did not pose a shielding effect during the e-beam irradiation. A graphite calorimetric dosimeter was used to accurately determine the dose absorbed. EPR qualitative measurements required irradiation at 77 K to allow thermal stabilization of the formed radicals for further observation and analysis. This was carried out using a ^60^Co gamma chamber at a dose rate of 2.4 kGy. After the absorption of the 10 kGy dose, EPR tests were conducted. Irradiation at 77 K using liquid nitrogen cannot be performed in an electron accelerator due to the danger of implosion of dewar necessary to ensure cryogenic conditions.

### 4.4. Immunohistochemical Stainings

The structure of the non-irradiated and irradiated hADMs was assessed by histochemical and immunohistochemical staining. The 6 mm biopsy punched lyophilized scaffold fragments were hydrated in PBS for 24 h. Next, the scaffold was fixed in 4% paraformaldehyde (PFA, Sigma Aldrich, St. Louis, MO, USA) for 24 h. Fixed specimens were paraffin-embedded using a tissue processor (Xpress Sakura, Sakura Finetek Poland, Warsaw, Poland). Then, 4 um scaffold microtome slices were placed on glass slides and automatically stained using a PTLink instrument. For the whole collagen structure visualization, Masson’s Trichrome Staining was used. For the detailed scaffold structure analysis, a panel of anti-collagen antibodies was used, namely: rabbit anti-human/mouse/rat collagen I (COL1A1) polyclonal antibody (Invitrogen, ThermoFisher, Waltham, MA, USA), rabbit anti-human collagen III polyclonal antibody (IgG; Abcam, Cambridge, UK), mouse anti-human collagen IV monoclonal antibody (IgG1, clone: COL94, ThermoFisher, Rockford, IL, USA), mouse anti-human collagen VII (COL7A1) monoclonal antibody (IgG1; clone:LH7.2, Thermofisher, Rockford, IL, USA), and rabbit anti-human collagen XVII (COL17A1) polyclonal antibody (IgG; Invitrogen; ThermoFisher, Rockford, IL, USA).

### 4.5. Collagen Release and Degradation Assay

Collagen fiber degradation and release from native (non-irradiated, 0 kGy) and irradiated scaffolds were assessed using the medium used for hADM hydration after 24 h incubation. Total protein from the medium was dissolved by the methanol/chloroform method. Briefly, 400 uL of methanol (Sigma-Aldrich, St. Louis, MO, USA) was added to 100 uL of protein sample. The suspension was mixed, and 100 uL of chloroform (Merck, Darmstadt, Germany) was added. Next, 300 uL of ultra-pure DNAse, RNAse free water was added (Qiagen,, Hilden, Germany) and mixed vigorously by vertexing, followed by centrifugation for 1 min in 14,000× *g*. The top aqueous layer was aspirated, and 400 uL of methanol was added to the protein flake (interphase) and chloroform (basal layer) and mixed. Next, the specimen was centrifuged for 5 min at 20,000× *g*. The supernatant was removed, and the pellet was dried on a heating block for 5 min. Finally, the pellet was resuspended in 4× Laemmli Sample Buffer (BioRad, Hercules, CA, USA) containing 2-Mercaptoethanol (Sigma-Aldrich, St. Louis, MO, USA) and heated to 95 °C for 5 min. on a heating block. The specimens were analyzed using a Western blot.

### 4.6. Pepsin Digestion and gel Electrophoresis of the Irradiated Human Skin Scaffolds

Irradiated samples were shredded, and 20 mg of each sample was added to 1 mL of 2 g/L pepsin from porcine stomach mucosa, 3200–4500 units/mg, (Sigma, St. Louis, MO, USA) in 3% acetic acid. The samples were then stirred for 72 h at 4 °C. The SDS-PAGE gel electrophoresis in non-reducing conditions was performed on ThermoFisher Bolt mini gels in MES SDS running buffer (Life, Carlsbad, CA USA) against the solution containing only pepsin. The SeeBlue prestained standard (Life, Carlsbad, CA, USA) was used as a mass control. The gels were visualized with Coomassie stain by a simple Coomassie stain/destain procedure (1 g Coomassie R250, 100 mL glacial acetic acid, 400 mL methanol, 500 mL H_2_O).

### 4.7. Western Blot

The specimens, both medium-derived and lysates, were subjected to electrophoresis on 4–12% Gel (BioRad). The electrophoretically separated proteins were transferred on a nitrocellulose membrane (BioRad, Hercules, CA, USA) using the TransBlot Turbo mixed molecular weight protocol. The membrane was washed three times for 5 min in washing buffer (10× concentrated PBS (Gibco, Paisley, UK) supplemented with 0.1% Tween 20 (SigmaAldrich, St. Louis, MO, USA). Next, membranes were blocked for one hour in blocking solution (5% non-fat skim milk in 0.1% Tween 20 (SigmaAldrich, St. Louis, CA, USA) supplemented 1× PBS (Corning, Manassas, VA, USA). Blocked membranes were washed three times and incubated overnight with primary antibodies for collagen I, collagen III, and collagen VII (as described previously). Subsequently, the membranes were washed three times and then incubated with specific HRP-conjugated antibodies for one hour. The blots were developed using the SuperSignal West Femto chemiluminescent substrate kit (Thermo Fischer Scientific, Rockford, IL, USA) and visualized on a ChemiDoc Touch system (BioRad).

### 4.8. EPR Spectroscopy

The EPR experiments were performed in the range from 100 K every 30 deg to 340 K. To achieve thermal equilibrium, the samples before measurement were kept for 5 min at each selected temperature. Two series of tests were carried out: at a microwave power of 10 mW or 0.1 mW. Briefly, scaffold fragments were positioned in the high-sensitivity resonant cavity of the EMXplus (Bruker, Karlsruhe, Germany) EPR spectrometer with the X-band. The ER 4131VT control system was used for thermal annealing of irradiated collagen to selected temperatures. The DPPH standard was used to calibrate the apparatus. Unless otherwise specified, EPR spectra were recorded using the following parameters: sweep width 25.0 mT, modulation amplitude 0.1 mT, resolution 2500 points, time constant 1.28 ms, conversion time 16 ms. Bruker’s software was used for the analysis of experimental spectra (WinEPR) and simulation of individual signals (Simfonia).

### 4.9. Gas Chromatography

The 10 mg samples of skin scaffolds, irradiated with the increasing doses of e-beam radiation, i.e., 0, 15, 25, and 35 kGy (Elektronika 10 MeV linear energy e-beam accelerator, IChTJ) were measured for the irradiation-induced hydrogen production and oxygen consumption. The samples were irradiated in the 3.5 mL glass vials closed with rubber stoppers and secured by the aluminum flip caps. The hydrogen and oxygen contents in the samples were measured on the gas chromatograph Shimadzu-2014 with a TCD-2014 detector. The 1 m long column packed with molecular sieves 5A was used. The data were acquired by the CHROMAX interface. The carrier gas was argon 99.99%, calibration gas hydrogen 99.99%. Operations were conducted with syringes vol. 10, 25, and 500 μL. The system was working at 35 °C, on the column kept at 40 °C and the detector at 100 °C. The rate of flow of carrier gas was 10 cm^3^/min. The values of the H_2_ production and O_2_ consumption rates were calculated from the linear regression coefficient values in the function of ionizing radiation energy (J).

### 4.10. PBMC Isolation and Stimulation

Healthy donor peripheral blood mononuclear cells (PBMCs) were isolated from the freshly obtained buffy coat by density gradient centrifugation as previously described. Briefly, a fresh buffy coat was diluted (1:10) with EDTA-supplemented PBS (PBS-EDTA). Next, blood was layered dropwise on a density gradient (Pancoll—PAN Biotech, Aidenbach, Germany), followed by centrifugation for 25 min at 1250× *g* at room temperature with low acceleration and without deceleration. Next, the interphase layer was acquired and washed in PBS-EDTA for 10 min at 300× *g*. The washing step was repeated nine times for platelet removal. Finally, two million freshly-isolated PBMCs were seeded in 24 well plates in cell culture media, namely RPMI1640 (PAN Biotech, Aidenbach, Gaermany), supplemented with 10% FBS, in the presence or absence of native (0 kGy) or irradiated 6 mm scaffold fragments. After a 5-day incubation, the supernatant was collected and stored at −80 °C for further cytokine assays.

### 4.11. Cytokine Assay

Concentrations of IFNγ, IL-17, IL-32, IL-1β, TNF (all from R&D Biosystems, Minneapolis, MN, USA), and IL-35 (Mybiosource, San Diego, CA, USA) were measured using commercially available ELISA tests as previously described [49,50]. Protein levels were analyzed with an automated light absorbance reader (LEDETEC96 system). The results were calculated according to the standard curve, generated by individual standard dilutions, by MicroWin 2000 Software.

### 4.12. Statistics

The statistical analysis of obtained continuous data was performed using GraphPad Prism 7 software (Graph-Pad Software). The nonparametric paired Wilcoxon test was used to compare changes between analyzed conditions. The differences were considered statistically significant at *p* < 0.05. The results are presented as a mean ± standard error of the mean (SEM).

## 5. Conclusions

In summary, we showed that radiation sterilization did not change the architecture and three-dimensional structure of hADM; however, it significantly affected both collagen fiber fragmentation and degradation, as well as the induction of free radicals in a dose-dependent manner. More importantly, the observed effects did not disturb the therapeutic potential of novel grafts, as indicated by the ELISA assay of cytokines. Therefore, we conclude that sterilization of novel abdominoplasty skin-derived hADMs using a dose of 35 kGy of gamma radiation ensures high dressing sterility with no significant changes in its structure and immunomodulatory properties.

## Figures and Tables

**Figure 1 ijms-22-08467-f001:**
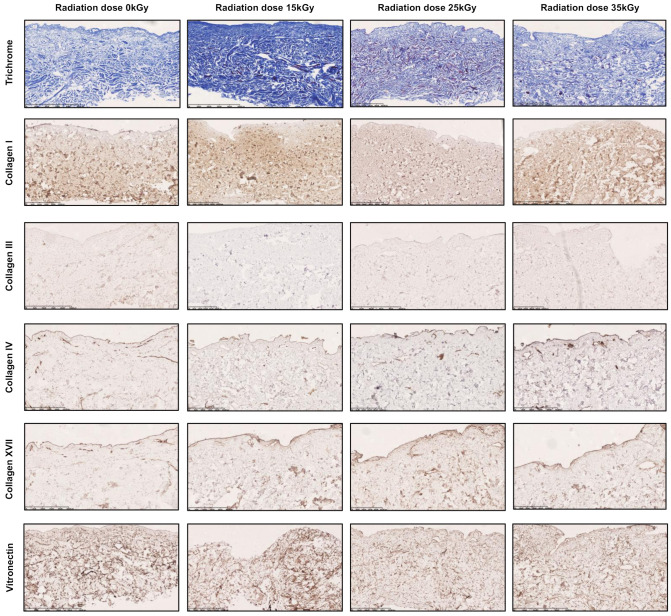
The effect of different doses of radiation sterilization on the collagen structure of decellularized abdominoplastic human dermal matrices. Representative histochemical (trichrome) and immunohistochemical (collagens I, III, IV, XVII, and vitronectin) staining of non-irradiated (0 kGy) and irradiated (15 kGy, 25 kGy, 35 kGy) novel human abdominoplastic skin-derived ADM. Scale bar for trichrome and vitronectin staining 500 µm; scale bar for collagen staining 250 µm.

**Figure 2 ijms-22-08467-f002:**
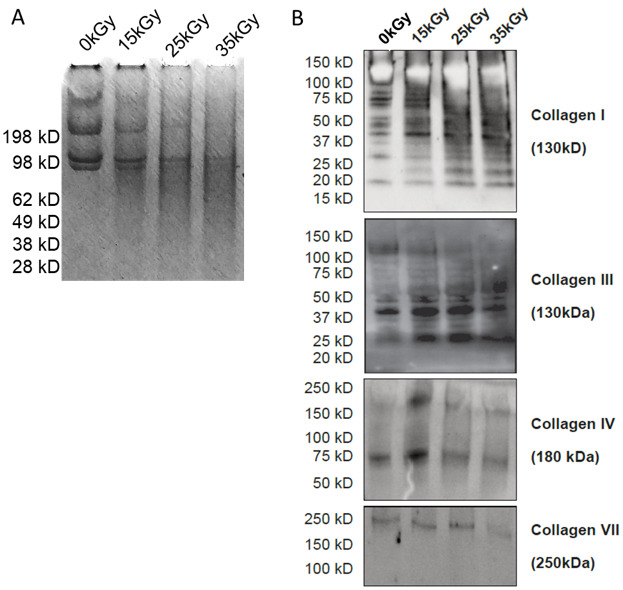
(**A**) Gel electrophoresis of the irradiated hADMs, digested with pepsin. From left to right: non-irradiated (0 kGy) control, and 15, 25, 35 kGy—irradiated samples. (**B**) Representative immunoblots from targeted analysis of collagen I, III, IV, and VII in hADM lysates.

**Figure 3 ijms-22-08467-f003:**
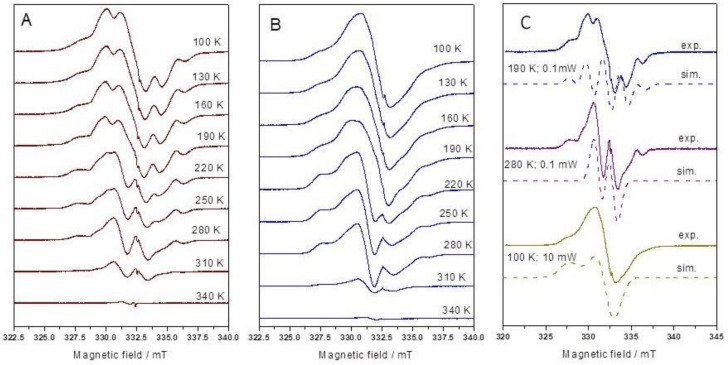
EPR spectra of human skin graft sterilized with a dose of 10 kGy at 77 K, then brought to 100 K and thermally annealed. The measurements were carried out at microwave power of (**A**) 0.1 mW and (**B**) 10 mW. (**C**) Experimental and simulated EPR spectra of hADM irradiated with a dose of 10 kGy at 77 K. Parameters used for simulations are given in the text.

**Figure 4 ijms-22-08467-f004:**
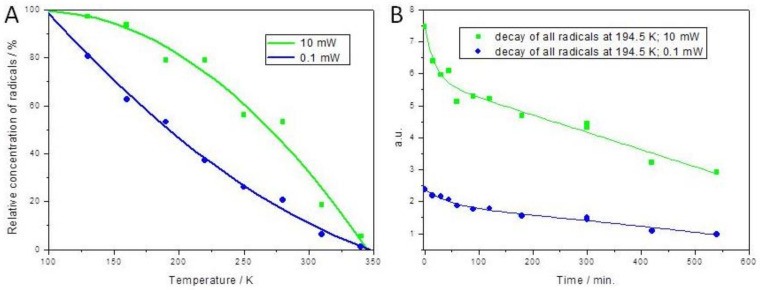
(**A**) The relative concentration of all radicals during annealing, calculated by double integration of spectra presented in Figure 3. The peroxyl radical signal is pronounced in the higher microwave power, as compared to alkyl radicals (pronouncedly seen in lower microwave power). (**B**) The decay of radicals in collagen irradiated and stored at the dry ice temperature calculated by double integration of the experimental spectra depicted in Figure 3.

**Figure 5 ijms-22-08467-f005:**
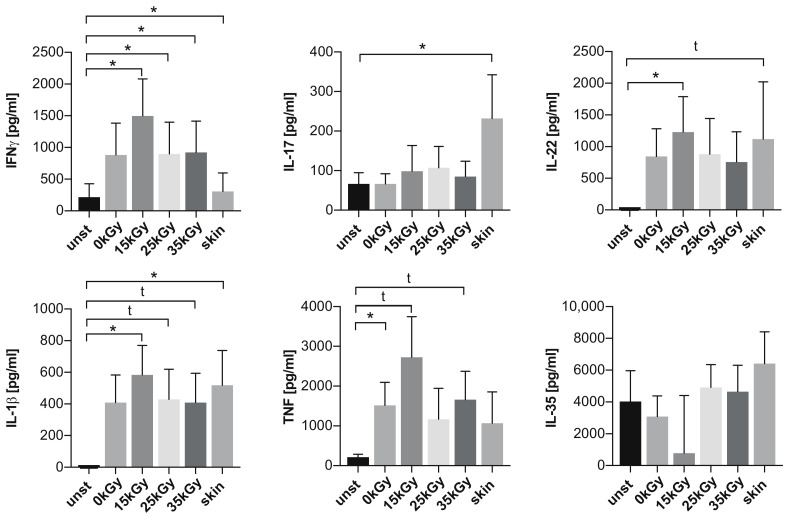
Summary of analyses of immunomodulatory properties of irradiated scaffolds. The levels of IFNγ, IL-17, IL-22, IL-1β, TNF, and IL-35 in cell culture supernatants from peripheral blood mononuclear cell alone (unst) or co-incubated with irradiated (15 kGy, 25 kGy, 35 kGy) hADM, non-irradiated (0 kGy) hADM, or unprocessed skin. *n* = 5; * *p* < 0.05; t trend at the threshold of statistical significance (0.1 > *p* > 0.05).

## Data Availability

The data presented in this study are available on request from the corresponding author. The data shared are in accordance with consent provided by participants on the use of confidential data.

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
