# Peer review of "Optimization of Novel Human Acellular Dermal Dressing Sterilization for Routine Use in Clinical Practice"

_ijms, 2021, doi:10.3390/ijms22168467_

Round 1

Reviewer 1 Report

The authors have studied the effect of sterilisation, by means of radiation, on acellular human dermis architecture and integrity. The authors suggest that this product could be used as a biological dressing to treat difficult to heal wounds and/or epidermolysis bullosa. A number of comments to make:

There are a number of sterile acellular human dermis products already described and available in the market. How this product will be different?

The authors claim in the abstract that the irradiation did not effect therapeutic potential of this product. How the authors drive this conclusion? There needs to be an in vivo testing to back up this claim.

Minor correction

Line 34- and EPR spectroscopy

Line 40- The last sentence in abstract has ben repeated twice.

Line 122- what features?

Line 203- Figure 5

Line 203- Discuss the significant differences observed for IFN-gamma, IL-1beta, IL-22 

Please discuss the effect of the amount of dry ice packed in thermo box. 

Reviewer 2 Report

The article evaluates the impact of sterilization on human acellular dermal dressing. They applied irradiation technique for sterilizing the ADM with different doses. The results demonstrated that irradiation causes degradation of collagen fibers and induced production of free radical. 

However, the immunological studies showed the modification of collagen didn’t disrupt the therapeutic application. This work provides a useful information regarding the sterilization process, but the data isn’t sufficient for the journal standard. The manuscript can be considered for 

publication after addressing the following concerns.

1. What is the novelty? Please clearly mention the significant of this work in the article.

2. In this work, the authors focused on collagen degradation during the sterilization process. However, there was another other structural proteins present in the ADM. Why the author could only focus on collagen? Please explain.

3. It is suggested to check the cell viability assay. The cell viability study will help to compare the sterilization doses effect on cell viability.

4. In the article authors mention cleavage of collagen occur during irradiation. It is preferable to include one more group regarding the ADM without sterilization which will ensure either decellularization process has impact or not collagen cleavage.

5. Please include the scale bar on Figure 1.

6. In conclusion authors mention “Therefore, we conclude 519 that sterilization of ADMs using a dose of 35kGy of gamma radiation ensures high dressing sterility while retaining its therapeutic potential to support effective wound healing”, There is no in-vivo data for wound healing in this article, Please modify.

Round 2

Reviewer 2 Report

The manuscript entitled "Optimization of novel human acellular dermal dressing sterilization for routine use in clinical practice" can be considered for publication.

Author Response

Dear Reviewer, thank you very much for your work. We are glad now the paper meets your requirements.

Kind regards,

Hanna Lewandowska, on behalf of co-authors.